# Rad51 filaments assembled in the absence of the complex formed by the Rad51 paralogs Rad55 and Rad57 are outcompeted by translesion DNA polymerases on UV-induced ssDNA gaps

**Laurent Maloisel◉\*, Emilie Ma, Jamie Phipps, Alice Deshayes, Stefano Mattarocci, Stéphane Marcand, Karine Dubrana, Eric Coïc◉\***

Université de Paris and Université Paris-Saclay, INSERM, CEA, Institut de Biologie François Jacob, UMR Stabilité Génétique Cellules Souches et Radiations, Fontenay-aux-Roses, France

\* laurent.maloisel@cea.fr (LM); eric.coic@cea.fr (EC)

## Abstract

The bypass of DNA lesions that block replicative polymerases during DNA replication relies on DNA damage tolerance pathways. The error-prone translesion synthesis (TLS) pathway depends on specialized DNA polymerases that incorporate nucleotides in front of base lesions, potentially inducing mutagenesis. Two error-free pathways can bypass the lesions: the template switching pathway, which uses the sister chromatid as a template, and the homologous recombination pathway (HR), which also can use the homologous chromosome as template. The balance between error-prone and error-free pathways controls the mutagenesis level. Therefore, it is crucial to precisely characterize factors that influence the pathway choice to better understand genetic stability at replication forks. In yeast, the complex formed by the Rad51 paralogs Rad55 and Rad57 promotes HR and template-switching at stalled replication forks. At DNA double-strand breaks (DSBs), this complex promotes Rad51 filament formation and stability, notably by counteracting the Srs2 anti-recombinase. To explore the role of the Rad55-Rad57 complex in error-free pathways, we monitored the genetic interactions between Rad55-Rad57, the translesion polymerases Polζ or Polη, and Srs2 following UV radiation that induces mostly single-strand DNA gaps. We found that the Rad55-Rad57 complex was involved in three ways. First, it protects Rad51 filaments from Srs2, as it does at DSBs. Second, it promotes Rad51 filament stability independently of Srs2. Finally, we observed that UV-induced HR is almost abolished in Rad55-Rad57 deficient cells, and is partially restored upon Polζ or Polη depletion. Hence, we propose that the Rad55-Rad57 complex is essential to promote Rad51 filament stability on single-strand DNA gaps, notably to counteract the error-prone TLS polymerases and mutagenesis.

**Data Availability Statement:** All relevant data are within the paper and its Supporting Information files.

**Funding:** This research was funded by Electricité de France (EDF), Commissariat à l'Energie Atomique et aux Energies Alternatives (CEA) Radiobiology call, Tandem Call CEA-PIC3i Institut Curie in Radiobiologie (to EC). The funders had no role in study design, data collection and analysis, decision to publish, or preparation of the manuscript.

**Competing interests:** The authors have declared that no competing interests exist.

## Author summary

Processive and accurate DNA polymerases replicate genomic DNA during the S phase of each cell cycle. DNA base lesions on template DNA block these polymerases and result in an accumulation of single-stranded DNA gaps behind moving replication forks. These gaps are filled-in by error-prone and error-free pathways. In this work, we show that the complex made by the Rad51 paralogs Rad55 and Rad57 is essential for the error-free homologous recombination gap-filling pathway when DNA replication is stalled by UV-induced DNA lesions, but not for DNA double strand break repair. Interestingly, we found that homologous recombination is efficiently outcompeted by error-prone translesion DNA polymerases in Rad55-Rad57-deficient cells. We propose that the Rad55-Rad57 complex is essential for Rad51 filament stability at UV-induced DNA gaps to promote efficient error-free homologous recombination. Furthermore, our study in yeast predicts that inhibitors of error-prone DNA polymerases might selectively target cancer cells in which RAD51 paralogs are mutated.

## Introduction

In all living organisms, genomic DNA undergoes chemical modifications or crosslinking with proteins. These damages greatly compromise DNA replication because they induce replication fork stalling. DNA damage tolerance mechanisms have evolved to ensure completion of genome replication [1], and rely on two main mechanisms: i) translesion synthesis (TLS) DNA polymerases and ii) the use of a homologous template, typically the newly synthesized sister chromatid. Specialized TLS polymerases efficiently insert nucleotides opposite and beyond lesions on DNA templates that block the replicative DNA polymerases [2]. They possibly extend blocked primer/template junctions at replication forks or at single-stranded DNA (ssDNA) gaps left behind the forks [3,4]. TLS polymerases are intrinsically error-prone and constitute a major source of DNA damage-induced mutagenesis [5,6]. Recombination-mediated pathways, such as the template switching and homologous recombination (HR) pathways, mediate damage bypass through the annealing of the damaged ssDNA gaps to the intact homologous template on the sister chromatid [7–9]. HR, also referred to as the salvage pathway, can use homologous chromosomes as intact donors rather than sister chromatids [10]. As TLS, template switching and HR compete for the same ssDNA substrates, they can partly compensate for each other [11–13]. However, template switching and HR are error-free lesion bypass mechanisms and counterbalance mutagenesis induced by TLS [14,15]. Thus, it is crucial to precisely characterize the factors that influence pathway choice to better understand genetic stability at replication forks.

The template switching pathway involves ubiquitin ligase and the Rad18/Rad6 pathway conjugating activities. In budding yeast, Rad6 and Rad18 induce PCNA mono-ubiquitination at its conserved K164 residue, whereas Rad5 and Mms2-Ubc13-dependent activities trigger its poly-ubiquitination at K164 [16,17] and the formation of X-shaped intermediates between sister chromatids when replication is challenged by DNA damages [8]. Interestingly, the formation of these X-shaped intermediates relies also on the HR factors Rad51 and Rad52 [7,18].

The HR pathway relies on the recombinase Rad51 that oligomerizes on ssDNA to form a right-handed helical nucleoprotein filament [19,20]. This filament performs homology search and catalyzes DNA joint formation between ssDNA and double-stranded DNA (dsDNA) homologous partners, thereby leading to strand exchange [21–23]. Eventually, repair DNA synthesis occurs from the damaged DNA invading ends on undamaged template homologous

DNA sequences [24]. Rad51 loading on RPA-coated ssDNA is a crucial step in HR and is mediated mainly by Rad52 in *Saccharomyces cerevisiae* and BRCA2 in metazoans [25–27]. Rad51 filament assembly also requires the activity of Rad51 paralogs (Rad55-Rad57 complex in *S. cerevisiae*; RAD51B, RAD51C, RAD51D, XRCC2, and XRCC3 in human cells; and the SHU complex in both [28]). Rad55 and Rad57 share 20% identity with the RecA/Rad51 ATPase core region [29,30]. However, they do not form filaments on ssDNA, and they do not exhibit strand exchange activity [31,32]. Electron microscopy images showed the association of gold-labeled Rad55 with Rad51 filaments assembled on ssDNA (Rad51-ssDNA) [32], and two-hybrid experiments revealed the interaction between Rad51 and Rad55 [33,34]. However, recent single-molecule studies suggest that the interaction between the Rad55-Rad57 complex and Rad51 filaments is transient and the Rad55-Rad57 complex dissociates during filament extension [35]. As similar findings were obtained in nematodes [36], it has been proposed that RAD51 paralogs behave as classical chaperones to temporarily assist Rad51 filament formation. More studies are required to clearly describe the precise role of each complex of Rad51 paralogs. In addition, each complex might play a specific role depending on the initial HR-inducing DNA lesion [37].

Rad55-Rad57 role in DNA double strand break (DSB) repair is considered accessory on the basis of the weaker sensitivity of *rad55* and *rad57* mutants to ionizing radiation and to site-directed DSBs compared with *rad51* mutants [38,39]. Interestingly, this sensitivity seems to depend on the Srs2 helicase activity because it is partially suppressed by *SRS2* ablation [32,39]. *In vitro* experiments have shown that the Srs2 translocase activity disrupts Rad51 filaments [40,41] and that the Rad55-Rad57 complex counteracts Srs2 to maintain Rad51 filaments on ssDNA [32,35]. Therefore, it has been proposed that Rad51 filament assembly and disassembly, which are mediated by the Rad55-Rad57 complex and Srs2 respectively, provide a regulatory mechanism to control HR initiation. However, *SRS2* deletion does not rescue spontaneous HR defects between direct repeats in the *rad55* and *rad57* mutants, indicating that the Rad55-Rad57 complex also acts independently of Srs2 [39].

To explain the different defects observed between spontaneous and DSB-induced HR, it was hypothesized that spontaneous HR between direct repeats is initiated by ssDNA gaps rather than DSBs [39]. Thus, the Rad55-Rad57 complex would play a more prominent role in HR when the initiating lesion is a ssDNA gap. In agreement, it has been observed that in cells lacking the TLS polymerases Pol ζ, *RAD55* ablation leads to a synergistic increase in DNA damage sensitivity [42–44].

Here, we explored Rad55-Rad57 role in Rad51 filament formation on ssDNA gaps and in the balance between HR and TLS in *S. cerevisiae*. For that purpose, we induced interhomolog HR in diploid strains by ionizing radiation (IR) or UV radiation. IR generates DSBs and ssDNA gaps [45], whereas UV generates mostly ssDNA gaps. UV-irradiated *S. cerevisiae* cells uncouple leading and lagging strand replication at irreparable UV lesions, thus generating long ssDNA regions on one side of the fork. Furthermore, small ssDNA gaps accumulate along replicated duplexes, likely resulting from repriming events downstream of the lesions on both leading and lagging strands. Translesion synthesis and homologous recombination counteract gap accumulation, without affecting fork progression [11]. Recently, RPA foci were used as a read-out of ssDNA gaps forming upon DNA replication through Methyl-methane sulfonate (MMS)- and UV-induced DNA damage [13]. They predominantly form far away from sites of ongoing replication, and they do not overlap with any of the repair centers associated with collapsed replication forks or DNA double-strand breaks. Instead, they represent sites of post-replicative DNA damage bypass involving translesion synthesis and homologous recombination. Therefore, we also used this mark to evaluate the cross-talk between TLS and HR.

We found that the Rad55-Rad57 complex is essential for UV-induced HR, but only accessory for IR-induced HR. Interestingly, this essential role is mainly Srs2-independent, and UV-induced HR in Rad55-Rad57-deficient cells can be restored by inactivation of a TLS polymerase (Polζ or Polη). Conversely, UV-induced HR cannot be restored upon inactivation of the template switch pathway in *MMS2* deficient cells. Overall, our results show that the Rad55-Rad57 complex is essential for Rad51 filament assembly on UV-induced ssDNA gaps. When this complex is absent, Rad51 filaments cannot prevent the recruitment of TLS polymerases and counterbalance mutagenesis.

## Results

### The Rad55-Rad57 heterodimer is essential for UV-induced homologous recombination

To investigate Rad55-Rad57 role in HR, we analyzed UV- and IR-induced interhomolog HR in wild type (WT) and *rad55Δ* isogenic diploid strains. The previous observation that *rad51* and *rad52* mutant cells are very sensitive to IR indicates that IR generates DSBs, whereas these mutants are much more resistant to UV radiation, indicating that DSBs are probably rarely induced [46,47]. Additionally, genetic evidence suggests that UV-induced HR is triggered by ssDNA gaps [48]. We measured interhomolog HR using two mutant alleles of *ARG4*: *arg4-RV*, a 2-bp deletion that ablates the *Eco*RV site at position +258, and *arg4-Bg*, a 4-bp insertion by fill-in of a *Bgl*II site at position +1,274 [49] (**Fig 1A**). These alleles do not revert [49] and only recombination in heteroallelic *arg4-RV/arg4-Bg* diploid cells results in the formation of a WT *ARG4* gene primarily by non-reciprocal transfer covering one mutation [50].

The *rad55Δ* diploid strain was not sensitive to UV radiation (**Fig 1B**), but remarkably, UV-induced recombinant [Arg+] frequency was strongly reduced in the *rad55Δ* diploid strain compared with WT cells (10-fold at 120 J/m$^2$) (**Fig 1C**). These results are very similar to those previously obtained with *rad51Δ* mutants [46], suggesting that the Rad55-Rad57 complex is determinant in UV-induced HR. Conversely, *rad55Δ* diploid cells were sensitive to IR (**Fig 1D**), but γ-ray-induced HR frequencies were identical in *rad55Δ* and WT cells at high doses (400 Gy and 600 Gy) (**Fig 1E**). We observed the same phenotypes in the *rad57Δ* mutant (**S1A–S1D Fig**). Likewise, it was previously reported that spontaneous HR rates are identical in *rad57Δ* and WT cells [38,51], a result we confirmed here in *rad55Δ* cells (**Fig 1F**). These observations suggest that the Rad55-Rad57 complex plays a specific and essential role in UV-induced HR, probably at ssDNA gaps. This complex is also involved in the repair of IR-induced DSBs, as shown by the lower survival of *rad55Δ* mutant. However, it is not essential since the IR-induced HR frequencies are comparable to WT.

### Resolution of UV-induced RPA foci is delayed in the *rad55Δ* mutant

To further examine the role of Rad55 in the management of ssDNA gaps induced by UV exposure, we monitored ssDNA by fluorescence microscopy using YFP-tagged Rfa1, the large subunit of the RPA complex [52]. It was indeed recently established that RPA foci formed upon UV-irradiation represent sites of post-replicative DNA damage bypass involving TLS and HR [13]. Exponentially growing cells were α-factor arrested in G1 and subjected or not to UV irradiation. In absence of irradiation, RPA foci were barely detectable in WT G1-arrested cells (**S1E Fig**). After UV-exposure, only a few G1-arrested WT cells displayed RPA foci (3.2% and 12.5% after 60 min and 240 min post-irradiation, respectively; **S1E Fig**). Unirradiated WT cells released into the cell cycle also exhibit few RPA foci (10% of the cells 60 min after release; **Fig 1H**). In marked contrast, 58% of WT irradiated cells displayed at least one RPA foci 60

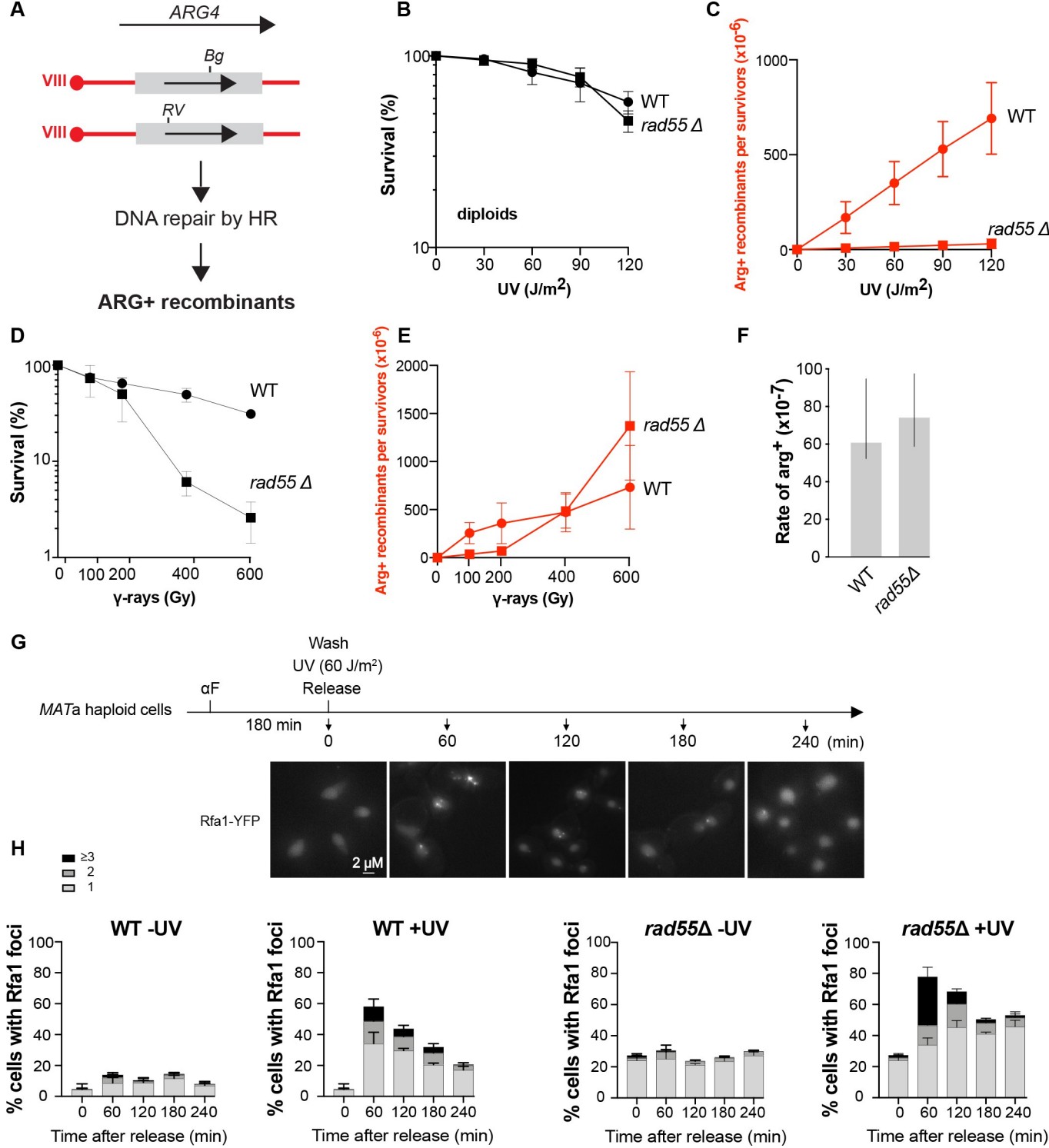

**Fig 1. Rad55 plays a major role in UV-induced HR.** (A) Schematic representation of the genetic system used. Frameshift mutations were introduced at the *Eco*RV site (RV, ±2 bp) or the *Bgl*II site (Bg, +4 bp). HR between the two *ARG4* mutant alleles can restore a WT *ARG4* allele associated with the [Arg+] phenotype. (B) Survival and (C) [Arg+] recombinant frequencies after UV exposure of WT and *rad55Δ* diploid cells. (D) Survival and (E) [Arg+] recombinant frequencies after γ irradiation of WT and *rad55Δ* diploid cells. (F) Spontaneous mitotic [Arg+] recombinant rates ([Arg+] x $10^{-7}$ /cell/generation) of WT and *rad55Δ* diploid strains. (G) Experimental scheme for microscopy analysis of RPA foci: Cells arrested in G1 phase with alpha-factor (αF) were exposed to UV before release into the cell cycle. Samples were collected for analysis every hour for four hours. Representative images of Rfa1-YFP cells are presented. Scale bars are 2 μm. (H) Quantification of Rfa1-YFP foci detected by fluorescence microscopy in WT and *rad55Δ* cells exposed or not to UV. Error bars indicate SDs from three independent experiments.

min after release in the cell cycle (**Fig 1H**). This confirms that UV-induced DNA damages induced in G1 leads to the formation of ssDNA gaps related to the encounter with DNA replication. Most of UV-induced RPA foci (70%) have been resolved 240 min after release (**Figs 1G and S1F**). This illustrates the efficient post-replicative DNA damage bypass in WT cells.

Surprisingly, a steady level of 20% of *rad55Δ* G1-arrested cells exhibits RPA foci (**S1E Fig**). This rate slowly increases after UV-exposure to reach 60% of G1-arrested cells at 240 min (**S1E Fig**). This newly described phenotype of *rad55Δ* cells is probably representative of an involvement of Rad55-Rad57 in the stabilization of repair intermediates of spontaneous and UV-induced DNA lesions in G1-arrested cells. 60 min after UV-exposure and release in the cell cycle, 80% of *rad55Δ* cells exhibit RPA foci (**Fig 1H**). Moreover, 31% of *rad55Δ* cells display three or more UV-induced RPA foci while this class is observed only in 10% of WT cells. In addition, contrary to WT, only 33% of RPA foci have been resolved 240 min post-irradiation. Finally, we observed an accumulation of *rad55Δ* cells in the second S phase 240 min after release from G1 arrest, probably related to the persistence of UV-induced gaps in absence of efficient HR (**S1F Fig**). Altogether, these results confirm that Rad55-Rad57 is important to manage ssDNA gaps forming upon UV-exposure.

In order to confirm that the different behavior of Rad55-Rad57 after UV- and IR-exposure is related to the difference in the lesions induced, we also observed RPA foci formation in WT and *rad55Δ* cells exposed to IR. To allow haploid cells to repair γ-induced DSBs by HR between sister chromatids, cells were irradiated 60 min after their release from the G1 arrest (**S2A Fig**). In WT and *rad55Δ* cells, we found the same proportion of cells displaying RPA foci after γ-rays exposure compared with UV, these foci being resolved with the same kinetics (**S2B Fig**). However, as already described in [13], 60 min after irradiation, 75% of the γ-irradiated cells with RPA foci display only one focus (**S2B Fig**), while this class represents 59% of UV-irradiated cells with RPA foci (**Fig 1G**). This difference could be attributed to the formation of a DSB repair center after γ-rays exposure while ssDNA gaps are not gathered after UV-irradiation. This difference is even more pronounced in the *rad55Δ* mutant. Indeed, 76% of γ-irradiated cells with RPA foci show only one focus while this class with a single focus represents 44% of UV-irradiated cells with RPA foci. These differences support our conclusion that Rad51 filament are assembled mainly on ssDNA gaps after UV exposure, while some events might be gathered in repair centers after γ-rays irradiation.

## UV-induced DNA lesions are channeled towards the *REV3* error-prone DNA repair pathway in the *rad55Δ* mutant

Although UV-induced HR was almost abolished in *rad55Δ* cells, these cells were resistant to UV radiation. We hypothesized that in *rad55Δ* mutant cells, UV-induced ssDNA gaps at replication forks could be repaired by TLS instead of HR. The TLS polymerase Polζ is required to bypass UV-induced DNA lesions, as illustrated by the strong UV-sensitivity of haploid cells lacking the Rev3 catalytic subunit of Polζ (**Fig 2A**) [53]. Conversely, *rev3Δ* diploid cells were much more resistant to UV radiation (**Fig 2B**), possibly due to HR promoted by *MAT* heterozygosity [51,54–56]. In addition, Polζ, is responsible for almost all UV-induced mutagenesis in yeast cells [53,57], as shown by the absence of UV-induced mutagenesis observed in the Polζ essential component *rev3* mutant [53]. Therefore, *rad55Δ* cells should strongly rely on Polζ and on its catalytic subunit Rev3 for UV resistance, and that UV-induced mutagenesis should increase in *rad55Δ* cells, as previously shown for spontaneous mutagenesis [42]. In agreement with previous studies, we confirmed that UV sensitivity is much higher in the *rev3Δ rad55Δ* double mutant than in the *rev3Δ* single mutant in haploid cells [42–44] (**Fig 2A**) and also in diploid cells (**Fig 2B**). This UV sensitivity is comparable with those we previously observed in

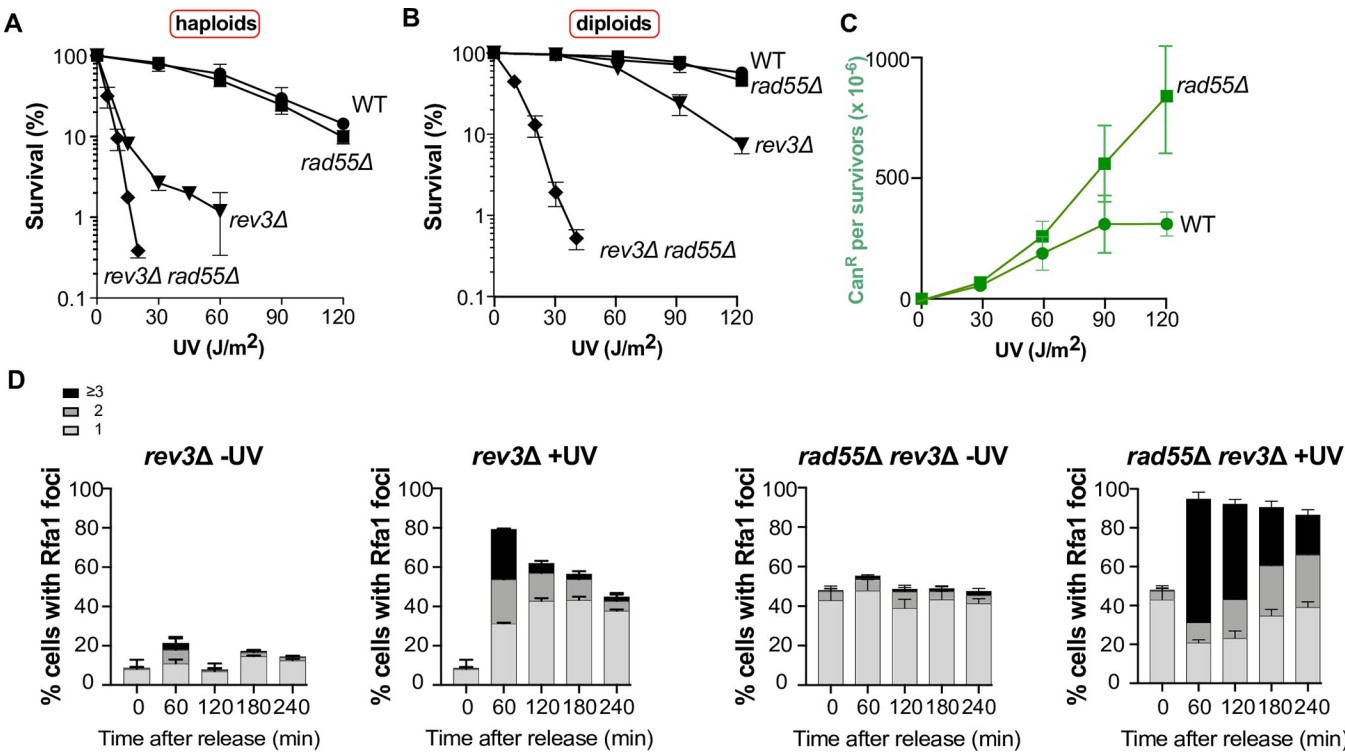

**Fig 2. Channeling UV-induced DNA lesions into the *REV3*-dependent pathway in *rad55Δ* cells.** (A) Survival of haploid cells following UV radiation. (B) Survival of diploid cells following UV radiation. (C) UV-induced mutagenesis in WT and *rad55Δ* cells determined by quantifying the Can[R] mutant frequency for the indicated UV doses. (D) Quantification of Rfa1-YFP foci detected by fluorescence microscopy in *rev3Δ* and *rad55Δ rev3Δ* cells not irradiated or after UV exposure. Error bars indicate SDs from three independent experiments.

the *rev3Δ rad51Δ* double mutant [58], confirming the essential role of Rad55-Rad57 in UV-induced HR. Therefore, the high resistance to UV of Rad55-deficient cells relies on the TLS pathway that could compensate for HR deficiency in these cells. Consequently, we expected an increase of mutagenesis frequency in *rad55Δ* cells. In agreement with this hypothesis, the frequency of UV-induced canavanine-resistant cells [59] was two-fold higher in the *rad55Δ* mutant than in WT cells (**Fig 2C**). This observation supports the channeling of DNA lesions towards the error-prone Polζ-dependent pathway in *rad55Δ* cells.

Measurement of RPA foci in UV-irradiated cells confirmed that TLS can compensate for HR. While the number of RPA foci was about the same in the *rev3Δ* mutant and in the *rad55Δ* mutant (**Figs 1G** and **2D**), the *rev3Δ rad55Δ* double mutant shows RPA foci in almost all cells with 64% of this population displaying at least three foci (**Fig 2D**). We also observed that *rev3Δ* mutants display an accumulation of cells in the second S phase 240 min after release from G1 arrest, while *rev3Δ rad55Δ* double mutant shows a block in the first G2 phase after the release (**S3 Fig**). The number of cells displaying RPA foci barely decreased with time in *rev3Δ rad55Δ* double mutant. However, the number of cells displaying at least three foci decreases strongly, which could represent repair events but also the disappearance of ssDNA gaps through degradation or breakage in the absence of repair. Considering the low survival of UV-irradiated *rev3Δ rad55Δ* cells, this more likely indicates that many ssDNA gaps formed during the first S phase after release cannot be repaired in absence of TLS and HR, leading to faulty mitosis. We also observed that the number of spontaneous RPA foci is 2-fold higher in the *rad55Δ rev3Δ* mutant compared with the *rad55Δ* mutant, indicating a large accumulation of ssDNA gaps

related to DNA replication independently of UV-exposure in this double-mutant (compare Figs 1G and 2D).

## UV-induced DNA lesions are channeled to the HR pathway in cells lacking TLS polymerases

The finding that UV-induced DNA lesions can be channeled to the TLS pathway in *rad55Δ* cells suggested that such lesions could be managed by the HR pathway in TLS-deficient cells. In that case, TLS-deficient cells should display a hyper-recombinogenic phenotype. To test this hypothesis, we measured the frequencies of UV-induced HR in *rev3Δ* (Polζ mutant) and *rad30Δ* (Polη mutant) cells. As expected, HR frequency was strongly increased in *rev3Δ* cells and to a lower extent in the *rad30Δ* mutant (**Fig 3A and 3B**). However, UV sensitivity was similar in diploid cells harboring the *rev3Δ* or *rad30Δ* mutation (**Figs 2B and 3C**). We propose that in the *rad30Δ* mutant, UV-induced DNA lesions can be bypassed by both HR and Polζ, while Polη cannot always take over Polζ function in *rev3Δ* cells, resulting in higher HR frequency. Additionally, it is important to note that accordingly with the strong sensitivity of *rev3Δ* cells exposed to UV, HR cannot compensate for all UV-induced DNA lesions that are supported by the TLS pathway.

## Translesion DNA polymerases prevent UV-induced HR in the *rad55Δ* mutant

We showed that UV-induced HR is strongly decreased in Rad55-Rad57 deficient cells. We also observed that the TLS and HR pathways compete for the same UV-induced ssDNA gaps, because mutagenesis was increased in *rad55Δ* mutants, and HR in *rev3Δ* mutants. Therefore, the recruitment of TLS polymerases at the lesion site might decrease HR in *rad55Δ* cells. We measured HR frequencies and observed a significantly increased frequency in the *rev3Δ rad55Δ* double mutant compared with *rad55Δ* cells (**Fig 4A**). This indicated that Polζ effectively impairs HR in the absence of the Rad55-Rad57 complex. We propose that the Rad55-Rad57 complex is important to stabilize Rad51 filaments on ssDNA gaps at the DNA lesion site, thus limiting the recruitment of TLS polymerases. Our results also suggest that in the absence of the Rad55-Rad57 complex, Rad51 filaments are functional enough to promote some HR events (**Fig 4A**), although this is not sufficient to improve UV resistance (**Fig 2**). We observed the same results when a Polη mutation (*rad30Δ*) was combined with *rad55Δ*

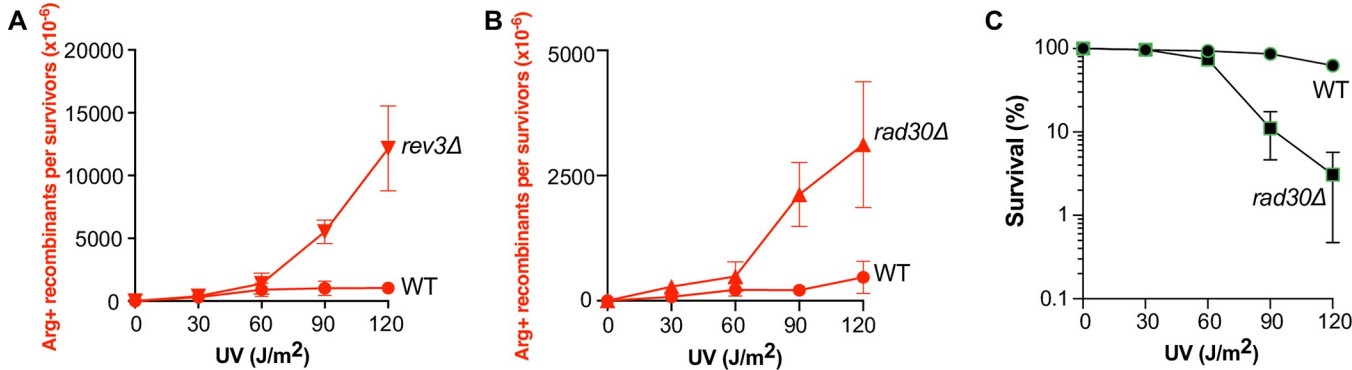

**Fig 3. Channeling UV-induced DNA lesions into the HR pathway in TLS-deficient cells.** (A) UV-induced [Arg+] recombinant frequencies in WT and *rev3Δ* diploid cells. (B) UV-induced [Arg+] recombinant frequencies in WT and *rad30Δ* diploid cells. (C) Survival of diploid *rad30Δ* cells after UV radiation.

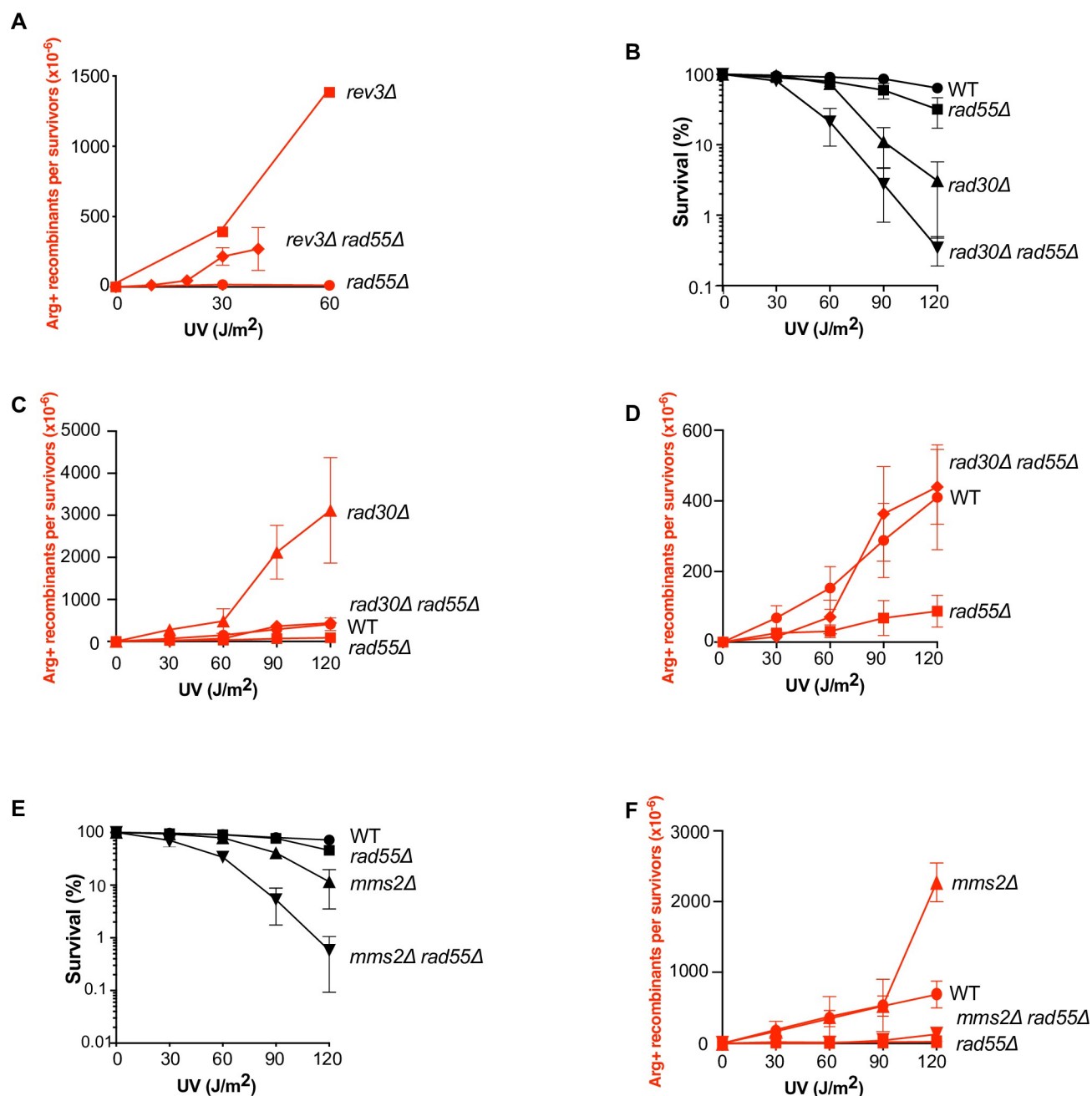

**Fig 4. *RAD55*-independent UV-induced HR in TLS-deficient cells.** (A) UV-induced [Arg+] recombinant frequencies in *rad55Δ rev3Δ* diploid cells. The very strong UV sensitivity displayed by the double mutant precluded the quantification of the UV-induced recombinant rate at UV doses higher than 40J/m². (B) Survival of *rad55Δ rad30Δ* diploid cells following UV radiation. (C) UV-induced [Arg+] recombinant frequencies in *rad55Δ* and *rad30Δ* diploids cells. (D) Close-up view of (C) to exclude the hyper-recombinogenic phenotype of *rad30Δ*. (E) *Rad55Δ mms2Δ* diploid cell-survival upon UV radiation. (F) UV-induced [Arg+] recombinant frequencies in *rad55Δ* and *mms2Δ* diploid cells.

(**Fig 4B–4D**). Thus, both Polζ and Polη contribute to the decrease of UV-induced HR in *rad55Δ* mutant cells.

Next, we asked whether UV-induced HR can be observed independently of the Rad55-Rad57 complex in cells deficient for the template switching pathway. Therefore, we measured the cell survival and recombination frequencies after UV radiation of cells in which

*MMS2* was deleted. We found that the *rad55Δ mms2Δ* double mutant was more sensitive than the single mutants (**Fig 4E**), suggesting that some template switching events can occur independently of HR. Strikingly, the frequency of UV-induced recombinants was as low in the *rad55Δ mms2Δ* double mutant as in the *rad55Δ* mutant (**Fig 4F**). This suggests that the template switching pathway does not compete with HR between homologous chromosome and that TLS polymerases still prevent HR in the *rad55Δ mms2Δ* double mutant. On the other hand, the recombination frequency in the *mms2Δ* single mutant was increased, but only after exposure to the highest UV dose (120 J/m$^2$, **Fig 4F**).

## The acute UV sensitivity of the *rad55Δ rev3Δ* double mutant is partially suppressed by *SRS2* deletion

As the Rad55-Rad57 complex limits Rad51 filament destabilization by Srs2 at DSBs [32,35], we asked whether Srs2 would be involved in the defective UV-induced HR associated with *rad55Δ* or *rad57Δ*. To address this question, we measured UV-induced HR frequencies in *rad55Δ srs2Δ* and *rad57Δ srs2Δ* double mutants. We found that they were similar to those observed in *rad55Δ* and *rad57Δ* single mutants (**Figs 5A and 5B, S4A, and S4B**). However, *rev3Δ rad55Δ srs2Δ* diploid cells were less sensitive to UV radiation than *rev3Δ rad55Δ* cells (**Fig 5C**). Thus, the Rad55-Rad57 complex is also involved in the protection of Rad51 filaments against Srs2 at ssDNA gaps formed after UV radiation, but this can only be seen in the absence of Polζ. When Polζ is active, the HR rate is probably too low in *rad55Δ* mutant cells to allow the detection of the effect of Rad51 displacement by Srs2.

Additionally, the level of resistance conferred by the deletion of *SRS2* in *rad55Δ rev3Δ* cells do not reach those of the *rev3Δ* single mutant (**Fig 5C**). This indicates that the Rad55-Rad57 complex plays a Srs2-independent role in UV-induced HR that is only revealed in *rev3Δ* mutants. We propose that the Rad55-Rad57 complex provides stability to Rad51 filaments independently of the protection against Srs2.

To sustain our conclusion, we analyzed UV-induced RPA foci in *srs2Δ* cells. The resolution of UV-induced RPA foci is delayed in *srs2Δ* cells compare to WT but in a lesser extent than in *rad55Δ* cells (**Figs 1G and 5E**). Moreover, the delay in UV-induced RPA foci resolution observed in the *rad55Δ* single mutant (**Figs 1G and S1F**) is the same than in the *rad55Δ srs2Δ* double mutant (**Figs 5F and S5**), suggesting that the antirecombinase activity of Srs2 does not account for the delay observed in the *rad55Δ* mutant. Similarly, the number of spontaneous and UV-induced RPA foci observed in *rev3Δ rad55Δ srs2Δ* haploid cells was the same as the number observed in the *rev3Δ rad55Δ* double mutant (**Fig 5F**). Moreover, the absence of Srs2 does not suppress the block in the first G2 phase after the release observed in the *rev3Δ rad55Δ* double mutant following UV irradiation (**S3 and S5 Figs**).

## The Rad55-Rad57 complex is essential for the formation of UV-induced lethal recombination events

We confirmed that the *srs2Δ* diploid strain is very sensitive to UV radiation as reported before [60]; however, this sensitivity was completely suppressed in *rad55Δ srs2Δ* and *rad57Δ srs2Δ* mutant cells (**Figs 5D and S4C**). Previous genetic studies led to the concept of toxic Rad51-dependent recombination intermediates that accumulate in the absence of Srs2 [61–63]. Therefore, on the basis of the complete suppression of *srs2Δ*-associated UV sensitivity by *rad55Δ* and *rad57Δ*, we propose that the Rad55-Rad57 complex participates in the formation of UV-induced Rad51 filaments that are toxic for the cells in the absence of Srs2. In agreement with previous reports [60,64,65], we observed a strong UV-induced hyper-recombination

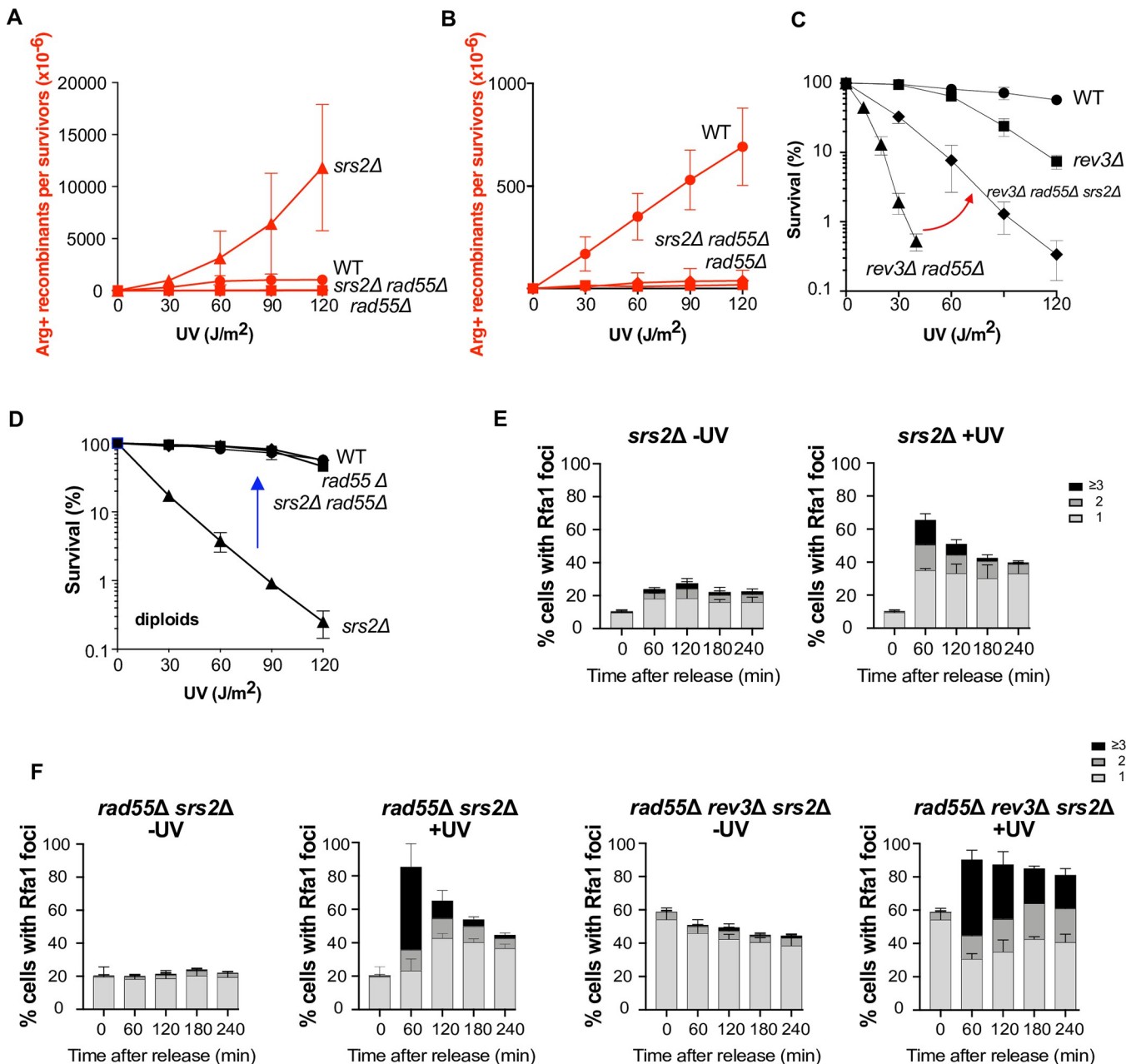

**Fig 5. Srs2 role in the *rad55Δ* mutant exposed to UV radiation.** (A) UV-induced [Arg+] recombinant frequencies in *rad55Δ srs2Δ* diploid cells. (B) Close-up view of (A) to exclude the hyper-recombinogenic phenotype of *srs2Δ*. (C) The acute UV sensitivity displayed by the *rad55Δ rev3Δ* mutant is partially suppressed by *SRS2* deletion. The red arrow between the *rev3Δ rad55Δ* and *rev3Δ rad55Δ srs2Δ* survival curves highlights the partial *srs2Δ* suppression. (D) Survival curves of diploid cells following UV radiation. The acute sensitivity to UV radiation of the diploid *srs2Δ* strain is suppressed by *rad55Δ* (blue arrow). (E) Quantification of Rfa1-YFP foci detected by fluorescence microscopy in *srs2Δ* cells not irradiated or after UV exposure. (F) Quantification of Rfa1-YFP foci detected by fluorescence microscopy in *rad55Δ srs2Δ* and *rad55Δ rev3Δ srs2Δ* cells not irradiated or after UV exposure. Error bars indicate SDs from three independent experiments.

phenotype in Srs2-deficient cells. This phenotype was completely suppressed by the concomitant deletion of Rad55 or Rad57 (**Figs 5A** and **S4A**).

### The translesion DNA polymerases Polζ and Polη are not essential for DSB repair

The large impact of the *rad55Δ rev3Δ* double mutant on cell survival after UV radiation suggested that the Rad55-Rad57 complex and Polζ play an important and specific role in UV-induced DNA repair. Although the Rad55-Rad57 complex is not essential for DSB repair [39], we wanted to determine the potential role of TLS polymerases in DSB repair in *rad55Δ* cells. To this aim, we measured the repair of a DSB induced at the *MAT* locus upon expression of the HO endonuclease controlled by a galactose-inducible promoter. The DSB was repaired by HR using an ectopic *MAT***a**-inc sequence inserted in chromosome V [65,66] (**Fig 6A**). After DSB induction, survival was decreased by 3-fold in *rad55Δ* cells compared with WT cells (**Fig 6B**). Conversely, survival of the *rev3Δ*, *rad30Δ*, and *rev3Δ rad30Δ* mutants was not affected by DSB induction, and they did not change the sensitivity of the *rad55Δ* mutant. This indicates that the TLS polymerases are not required for DSB repair efficiency.

## Discussion

### The Rad55-Rad57 complex is essential for HR-mediated repair of UV-induced ssDNA gaps

Genetic studies provided evidence that ssDNA gaps are the major initiator of spontaneous and UV-induced HR in yeast [47,57]. In addition, electron microscopy and two-dimensional gel electrophoresis showed that UV-irradiated cells accumulate ssDNA gaps, likely resulting from re-priming events downstream of stalled replication forks at UV-induced DNA lesions [11]. UV-induced ssDNA gaps were also inferred from a physical assay [12] and from the study of RPA foci distribution relatively to sites of ongoing replication [13]. Interestingly, both studies reported that TLS and HR counteract ssDNA gap accumulation.

Here, we show that the complex formed by the Rad51 paralogs Rad55 and Rad57 is essential for UV-induced ssDNA gaps repair, while accessory at DSB sites. We observed a strong decrease of HR frequencies in the *rad55Δ* and *rad57Δ* mutants specifically after UV radiation. Moreover, we confirmed the synergistic increase of UV sensitivity upon depletion of both Rad55 and Polζ in haploid and diploid cells (**Fig 2A and 2B**). This was confirmed by the large

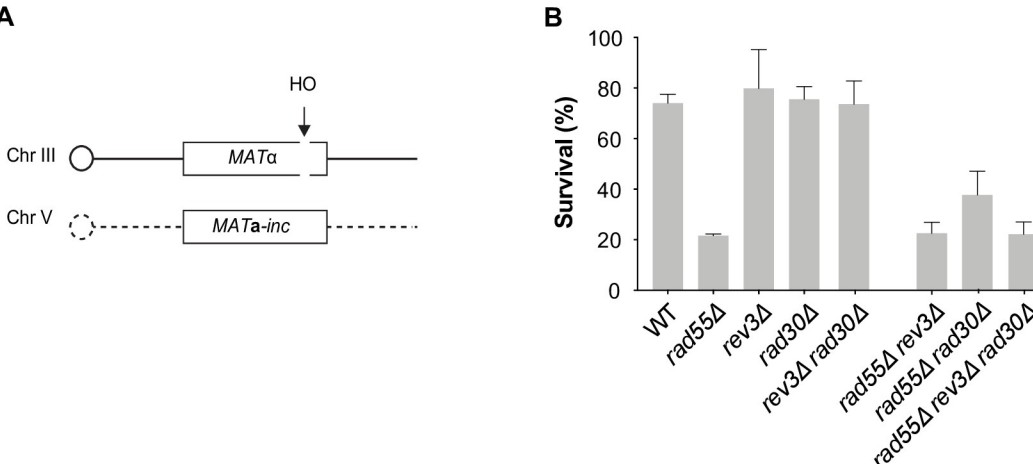

**Fig 6. TLS DNA polymerases are not required for cell survival after a site-specific DSB.** (A) Schematic representation of the system used to measure repair of a HO-induced DSB by gene conversion between ectopic copies of *MAT*. (B) Cell viability after DSB formation.

increase in the number of UV-induced RPA foci observed in the *rad55Δ rev3Δ* double mutant (**Fig 2D**). We did not observe this negative interaction after induction of a site-directed DSB (**Fig 6**). In addition, we found that UV-induced mutagenesis at the *CAN1* locus was increased in the *rad55Δ* mutant (**Fig 2C**), and that UV-induced HR was increased in Pol ζ and Pol η deficient cells (**Fig 3A and 3B**). Most importantly, we observed UV-induced HR in the absence of the Rad55-Rad57 complex only in cells deficient for TLS (**Fig 4**). These data suggest that the Rad55-Rad57 complex might stabilize Rad51 filaments on ssDNA gaps to prevent the recruitment of the TLS polymerases.

## The Rad55-Rad57 complex plays three prominent roles in UV-induced HR

One of the best defined role of Rad55-Rad57 is the protection of Rad51 filaments from Srs2 at DSBs [32,35]. Here, we found that although the defect in UV-induced HR observed in *rad55Δ* cells was not suppressed by *SRS2* deletion (**Fig 5A and 5B**), the very high UV sensitivity of the *rad55Δ rev3Δ* double mutant was partially suppressed by *srs2Δ* (**Fig 5C**). This suggests that the Rad55-Rad57 complex protects Rad51 filaments against Srs2 also at UV-induced ssDNA gaps, although Esc2- and Elg1-dependent mechanisms also regulate negatively Srs2 at sites of stalled replication forks [67,68]. Additional studies are required to determine the interplay between these regulating factors.

The increased UV-sensitivity of the *rev3Δ rad55Δ srs2Δ* triple mutant compared with the *rev3Δ* single mutant (**Fig 5C**) clearly indicates that the Rad55-Rad57 complex plays a Srs2-independent role in UV-induced HR. We propose that the second role of this complex is to provide stability to Rad51 filaments, independently of the protection against Srs2. In agreement with this conclusion, we found that in *rad55Δ* or *rad57Δ* mutants, *srs2Δ*-associated UV-sensitivity was completely suppressed (**Fig 5D** and **S4C**). This indicates that in the absence of Srs2 activity, Rad51 stabilization by the Rad55-Rad57 complex would lead to lethal events, possibly initiated by ssDNA gaps that could block replication fork restart. On the basis of the results of single-molecule studies, it was proposed that the Rad55-Rad57 complex helps Rad51 filament formation by acting as a chaperone [35], in agreement with our observations.

Remarkably, UV-induced HR frequency increased in *rad55Δ* cells upon Polζ or Polη deletion (**Fig 4**). Therefore, the third role of the Rad55-Rad57 complex would be to allow HR to outcompete the TLS polymerases. Rad51 filament stabilization by the Rad55-Rad57 complex on ssDNA at the lesion site could prevent the recruitment of TLS polymerases by PCNA because of structural constraints. In the absence of Rad55-Rad57, unstable Rad51 filaments cannot prevent the loading of the TLS polymerases that inhibit HR and induce mutagenesis. TLS polymerase depletion in *rad55Δ* mutants would allow some HR events to occur, but the inherent instability of Rad51 filaments would make them rare and explain the low survival rate of *rev3Δ rad55Δ* mutant cells. Our results clearly highlight that UV-induced lesions can be channeled from HR to TLS and *vice versa*, when one pathway is inactivated. This provides further support to a model in which HR and TLS can share common substrates [12,13].

## Model for the activity of the Rad55-Rad57 complex in ssDNA gap repair

Our genetic data suggest three different and essential functions for the Rad55-Rad57 complex in UV-induced interhomolog HR initiated at ssDNA gaps (**Fig 7A**). First, this complex is required for Rad51 filament formation and stabilization that are essential for efficient strand exchange at ssDNA gaps. The 5'ends of DSBs are resected to generate 3' ssDNA that can invade the homologous donor. Conversely, 3' ssDNA is not directly available at ssDNA gaps. Therefore, the Rad55-Rad57 complex may be required to form Rad51 filaments that can invade without 3' ssDNA extremities. Alternatively, the Rad55-Rad57 complex can be involved

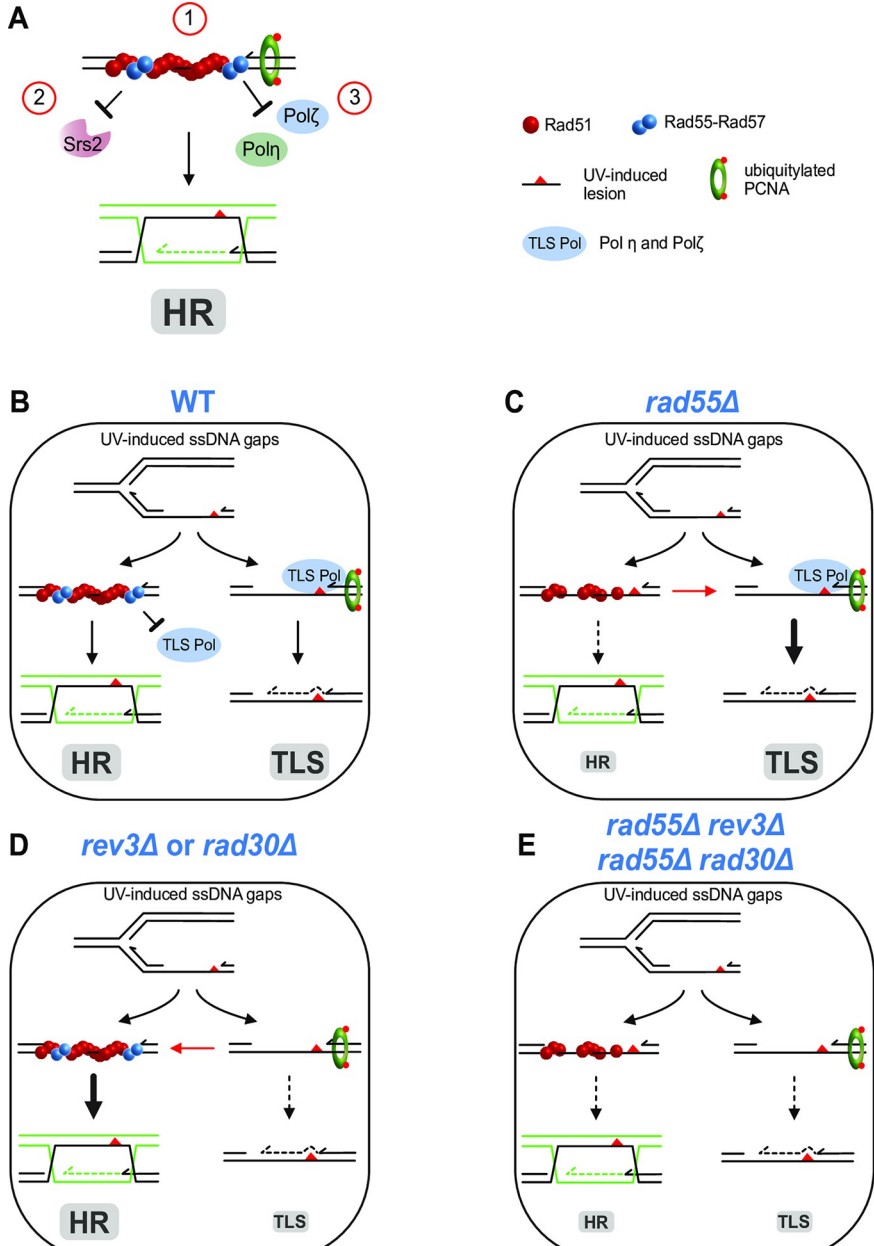

**Fig 7. Model of Rad55-Rad57 roles in the repair of ssDNA gaps by HR behind replication forks.** The figure shows an ssDNA gap generated after stalling of a replicative DNA polymerase at a UV-induced DNA lesion. This gap is a substrate for both the HR and TLS pathways. (**A**) The three roles of the Rad55-Rad57 complex in Rad51 filament formation on ssDNA gaps are highlighted: (1) formation of functional Rad51 filaments; (2) protection against Srs2 dismantling activity; (3) prevention of TLS polymerase recruitment at the primer/template junction. (**B**) In WT cells, Rad55-Rad57 heterodimers form and stabilize Rad51 filaments on ssDNA leading to HR, avoiding bypass by TLS. Rad51 filaments could prevent the recruitment of the TLS polymerases Polζ and Polη (collectively named TLS Pol) by covering the 3' extremity blocked at the DNA lesion. (**C**) In the *rad55Δ* mutant, inefficient and/or fewer Rad51 filaments are formed that cannot prevent the recruitment of the TLS polymerases. Therefore, more UV-induced DNA lesions are channeled towards the TLS pathway (bold arrow). (**D**) In the *rev3Δ* or in the *rad30Δ* mutant, the absence of Polζ or Polη allows channeling UV-induced DNA lesions towards the HR pathway (bold arrow). (**E**) In the *rad55Δ rev3Δ* and *rad55Δ rad30Δ* double mutants both the HR and the TLS pathways are impaired.

in the 3' end denaturation at the lesion site to generate a 3' ended ssDNA by recruitment of a DNA helicase. Second, the Rad55-Rad57 complex within Rad51 filaments counterbalances Srs2 activity through a mechanism that remains poorly understood [32,35]. Third, the Rad55-Rad57 complex could inhibit TLS polymerases by limiting their loading at the primer/template junction within ssDNA gaps, probably through the stabilization of Rad51 filaments (**Fig 7B-7E**). This competition might be crucial to control mutagenesis resulting from gap-filling by Polζ, but further studies will be required to understand how the access of HR and TLS are regulated at the site of lesions and to determine the nature of the lesions that can only be managed by TLS, as shown by the acute sensitivity of UV-irradiated *rev3Δ* mutant. Thus, in our model, Rad51 filaments formed with the Rad55-Rad57 complex preserve the genome stability through ssDNA gap repair by HR, but also through competition with TLS on common ssDNA gap substrates.

### The interplay between HR and TLS in mammals

The interplay between error-prone and error-free lesion-bypass pathways is documented in *E. coli* and yeast (the present study, [13,14]), and it has been recently investigated in mammals. First of all, it was recently shown that BRCA-deficient tumors sensibility to cisplatin and synthetic lethality with PARP inhibitors were associated with a BRCA default in preventing replication-associated gaps rather than in preventing DSBs [69,70]. More specifically, HR can be triggered by PRIMPOL-dependent ssDNA gaps behind stalled replication forks [71]. Moreover, BRCA1- and BRCA2-defective breast cancer cells, display an increased somatic mutational load specific of TLS polymerases [72–75]. Accordingly, a recent study found that BRCA1/2-defective cancer cells rely much more on TLS for repair of PRIMPOL-dependent ssDNA gaps [76]. In addition, similarly to PARP1 or DNA polymerase Polθ inhibitors that are used to treat HR-deficient cancers [77–79], the small molecule JE-RH-06 disrupts the interaction between the TLS polymerases REV1 and Polζ and shows preferential cytotoxicity in BRCA1-deficient cancer cells [76,80]. Thus, it would be interesting to determine whether the association of cancer-associated RAD51 paralog mutations and TLS mutations leads to a synergic sensitivity to replication fork blocking lesions, as we observed in yeast in the *rad55Δ rev3Δ* double mutant. If an acute sensitivity is found, one might expect a cytotoxicity associated with TLS polymerase inhibitors in RAD51 paralog-deficient cancer cells.

## Materials and methods

### Yeast strains, plasmids and media

The *S. cerevisiae* strains used in the present study are isogenic derivatives of FF18733 (*his7-2, leu2-3,112, lys1-1, trp1-289, ura3-52*), JKM146 (*hml::ADE1 MATalpha hmr::ADE1, arg5,6:: MATa-inc::HPH1, ade3::GAL::HO*) and W303 (*ADE2 leu2-3,112 his3-11,15 trp1-1, ura3-1*) and are listed in **S1 Table**. Yeast cells were incubated at 30˚C for all the experiments described. Gene deletions were performed by PCR-mediated one-step replacement [81,82]. Mutants were selected on YPD medium containing 300 mg/L geneticin (Sigma) or nourseothricin (clo-NAT; Werner BioAgents). All deletions were confirmed by PCR amplification of genomic DNA. All media were prepared as previously described [83].

### Irradiation, cell survival assay, and measurement of DNA damage-induced HR frequency

Cells in stationary phase and in exponential growth phase were used for irradiation with UV and γ-rays, respectively. UV irradiation was performed using a 264 nm source. Yeast cells were irradiated with 30, 60, 90 and 120 J/m$^2$ (2 J/m$^2$/s dose rate). γ-irradiation was performed using

a GSR D1 irradiator (Gamma-Service Medical GmbH). It is a self-shielded irradiator with four sources of $^{137}$Cesium. The total activity was 180.28 TBq in March 2014. As yeast cells resuspended in 1 mL of sterile $H_2O$ were irradiated in 1.5ml plastic microtubes, dosimetry was performed using plastic microtubes with a cylindrical ionizing chamber 31010 (PTW, Freiburg, Germany) following the American Association of Physicists in Medicine recommendations [84]. This ionizing chamber has a cavity of 0.125 cm$^3$ calibrated with $^{137}$Cesium in air kerma free in air and the reference number of our facility is 210382401. The polarity and the ion recombination were measured for this $^{137}$Cesium source. Each measurement was corrected with the KTP factor to take into account the variations in temperature and atmospheric pressure. Yeast cells were irradiated at 100, 200, 400 and 600 Gy (single doses) and with a 12 Gy/min dose rate that takes the radioactive decrease into account.

Before (UV) or after (γ-rays) irradiation, cells were plated at the appropriate dilution on rich medium (YPD) to measure the survival rate, and on synthetic plates without arginine to quantify the number of HR events between *ARG4* heteroalleles. HR frequencies were determined by dividing the number of recombinant colonies growing on selective medium by the number of unselected colonies subjected to the same dose of irradiation. The values obtained were corrected by subtracting the number of recombinants on the non-irradiated plates. The mean percentage from at least three independent experiments is presented.

## Measurement of spontaneous heteroallelic HR

Rates of spontaneous HR between two heteroalleles of *ARG4* were determined by fluctuation tests using the method of the median [85]. The reported rates were obtained from three independent experiments, each performed with nine independent 2-ml cultures started with less than 200 cells and incubated at 30˚C for three days. The significance of the rates is indicated by the 95% confidence interval [85].

## Measurement of UV-induced mutagenesis

UV-induced mutagenesis was measured with the *CAN1* forward-mutation assay [59]. UV-induced mutagenesis frequencies were obtained by dividing the number of colonies growing on synthetic plates without arginine and containing l-canavanine (Sigma, 30 mg/l) (*i.e.*, canavanine-resistant, Can$^R$, cells) by the number of cells that survived irradiation counted on rich media YPD. The number of Can$^R$ colonies obtained after irradiation was corrected by subtracting the number of spontaneous Can$^R$ colonies growing on non-irradiated plates.

## Cell growth, synchronization and irradiation before microscopy

Cells were grown at 30˚C in synthetic complete medium (Formedium). Before UV- and γ-irradiation, growing cells were synchronized in G1 using 17 mg/ml α-factor for 180 min. They were released from the G1 arrest by three consecutive washes with sterile $H_2O$. Cells were resuspended in 50 ml of sterile $H_2O$ and UV-irradiated in a glass plate of diameter 190 mm. UV irradiation was performed using a 264 nm source. Yeast cells in water were irradiated with 60 J/m2 (2 J/m2/s dose rate) then resuspended in fresh complete medium. γ-irradiation was performed as described for survival studies. Cells were then resuspended in fresh complete medium. Time-course experiment were started with yeast cultures adjusted to $OD_{600}$ 0.5.

## Microscopy

Live-cell images were acquired using a wide-field inverted microscope (Leica DMI-6000B) equipped with Adaptive Focus Control to eliminate Z drift, a 100×/1.4 NA immersion

objective with a Prior NanoScanZ Nanopositioning Piezo Z Stage System, a CMOS camera (ORCA-Flash4.0; Hamamatsu) and a solid-state light source (SpectraX, Lumencore). The system is piloted by MetaMorph software (Molecular Device).

YFP images were acquired at indicated time points after alpha factor block release and UV or γ-rays irradiation; 19 focal steps of 0.20 μm were acquired with an exposure time of 100 ms using a solid-state 500-nm diode and a YFP filter (excitation 470–510 nm and dichroic 495 nm; Chroma Technology Corp.). A single bright-field image on one focal plane was acquired at each time point with an exposure of 50ms. All the images shown are z projections of z-stack images. Image analysis was achieved following processing with ImageJ software (National Institutes of Health), using scripts written in ImageJ macro language. Bright-field and maximum intensity projections of YFP images were merged, Rfa1-YPD foci were then quantified in 2D.

## Measure of DNA content by flow cytometry

1 ml of cells were resuspended in 70% Ethanol and kept at 4˚C. Cells were centrifuged for 3 min at 7,000 RPM and washed once in PBS, resuspended in PBS with RNase A (0.5 mg/ml) for 2 hours at 50˚C. Cells were centrifugated and resuspended in PBS with iodure propidium (50 μg/ml) for one hour at room temperature. Cells were diluted 10-fold in PBS and cell aggregates were dissociated by sonication. DNA content analysis was performed using LSRII flow cytometer (BD Biosciences) and analyzed with the FlowJo software.

## Survival following DSB formation

Cells were grown overnight in liquid culture medium containing lactate acid instead of glucose. Survival following HO-induced DSB was measured as the number of cells growing on galactose-containing medium divided by the number of colonies growing on YPD. The results shown are the mean values of at least 3 independent experiments.

## Supporting information

**S1 Fig. Rad57 plays a major role specifically in UV-induced HR.** (A) Survival, and (B) [Arg +] recombinant frequencies in WT and *rad57Δ* diploid cells following UV radiation. (C) Survival and (D) [Arg+] recombinant frequencies in WT and *rad57Δ* diploid cells after γ irradiation. (E) Quantification of Rfa1-YFP foci in WT and *rad55Δ* cells not irradiated or after UV exposure in G1-arrested cells. Error bars indicate SDs from three independent experiments. (F) Representative images of Rfa1-YFP WT and r*ad55Δ* cells released from G1 arrest and UV-irradiated or not. Bright-field (BF) images are merged with YFP images. Scale bars are 2 μm. FACS profiles for each corresponding time point are shown.
(TIF)

**S2 Fig. γ-rays-induced RPA foci in WT and *rad55Δ* strains.** (A) Experimental scheme: Cells arrested in G1 phase with alpha-factor (αF) were release into the cell cycle. After one hour from the release, cells were exposed to γ-rays. Samples were collected every hour for four hours. (B) Quantification of Rfa1-YFP foci in WT and *rad55Δ* cells not irradiated or after γ-rays irradiation. Error bars indicate SDs from three independent experiments.
(TIF)

**S3 Fig. UV-induced RPA foci in the *rev3Δ* and *rad55Δ rev3Δ* strains.** Representative images of Rfa1-YFP *rev3Δ* and *rad55Δ rev3Δ* cells after release from G1 arrest and UV irradiated or not. Bright-field (BF) images are merged with YFP images. Scale bars are 2 μm. FACS profiles for each corresponding time point are shown.
(TIF)

**S4 Fig. Srs2 roles in the *rad57Δ* mutant exposed to UV radiation.** (A) UV-induced [Arg+] recombinant frequencies in *rad57Δ srs2Δ* diploid cells. (B) Close-up view of (A) to exclude the hyper-recombinogenic phenotype of *srs2Δ*. (C) Survival curves of diploid cells following UV radiation. The acute sensitivity to UV radiation of the diploid *srs2Δ* strain is suppressed by *rad57Δ* (blue arrow).
(TIF)

**S5 Fig. RPA foci in *rad55Δ rev3Δ srs2Δ* cells.** Representative images of Rfa1-YFP *srs2Δ*, *rev3Δ srs2Δ* and *rad55Δ rev3Δ srs2Δ* cells after release from G1 arrest and UV irradiated or not. Bright-field (BF) images are merged with YFP images. Scale bars are 2 μm. FACS profiles for each corresponding time point are shown.
(TIF)

**S1 Table. *Saccharomyces cerevisiae* strains.**
(PDF)

## Acknowledgments

We thank Jim Haber and Pablo Radicella, for critical and careful reading of the manuscript. We thank Véronique Ménard from the radiation facility of our institute for her help with the use of the GSRD1 irradiator. We also appreciate the help of Elisabetta Andermarcher with the English editing.

## Author Contributions

**Conceptualization:** Laurent Maloisel, Eric Coïc.

**Formal analysis:** Laurent Maloisel.

**Funding acquisition:** Eric Coïc.

**Investigation:** Laurent Maloisel, Emilie Ma, Jamie Phipps, Alice Deshayes, Stefano Mattarocci, Eric Coïc.

**Methodology:** Laurent Maloisel, Emilie Ma, Alice Deshayes, Stéphane Marcand, Karine Dubrana, Eric Coïc.

**Project administration:** Eric Coïc.

**Resources:** Laurent Maloisel, Emilie Ma, Eric Coïc.

**Software:** Jamie Phipps.

**Supervision:** Stéphane Marcand, Karine Dubrana, Eric Coïc.

**Validation:** Laurent Maloisel, Emilie Ma, Alice Deshayes, Stefano Mattarocci, Eric Coïc.

**Visualization:** Laurent Maloisel.

**Writing – original draft:** Laurent Maloisel, Eric Coïc.

**Writing – review & editing:** Laurent Maloisel, Jamie Phipps, Stéphane Marcand, Karine Dubrana, Eric Coïc.

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
