## [Decision Letter · Decision Letter 0]

11 Apr 2022

Dear Dr Coic,

Thank you very much for submitting your Research Article entitled 'Rad51 filaments assembled in the absence of the complex formed by the Rad51 paralogs Rad55 and Rad57 are outcompeted by translesion DNA polymerases on UV-induced ssDNA gaps' to PLOS Genetics.

The manuscript was fully evaluated at the editorial level and by independent peer reviewers. The reviewers appreciated the attention to an important problem, but raised some concerns about the current manuscript. In particular, two reviewers requested physical evidence for Rad51 filament formation at ssDNA gaps. This issue could potentially be addressed by Rad51 fluorescence microscopy or by assessing Rad51 bound to the chromatin fraction. Based on the reviews, we will not be able to accept this version of the manuscript, but we would be willing to review a much-revised version. We cannot, of course, promise publication at that time.

If you decide to revise the manuscript for further consideration at PLOS Genetics, please aim to resubmit within the next 60 days, unless it will take extra time to address the concerns of the reviewers, in which case we would appreciate an expected resubmission date by email to plosgenetics@plos.org.

[LINK]

We are sorry that we cannot be more positive about your manuscript at this stage. Please do not hesitate to contact us if you have any concerns or questions.

Yours sincerely,

Lorraine S. Symington

Associate Editor

PLOS Genetics

Gregory P. Copenhaver

Editor-in-Chief

PLOS Genetics

Reviewer's Responses to Questions

**Comments to the Authors:**

Reviewer #1: In this pithy manuscript by Maloisel et al, the authors examined the role of Rad55 in UV-induced recombination. They showed that despite not being UV sensitive, rad55∆ cells are unable to perform allelic recombination upon UV damage. This is due to TLS bypassing these lesions in an error-prone manner. The authors argue that genetic relationship between RAD55 with REV3 and RAD30 is related to gap repair and not DSB repair. The novelty of this study is the analysis of UV-induced damage and the finding that HR is needed to bypass UV-induced gaps and not direct DSBs. A few experiments detailed below would help provide more direct evidence and mechanistic understanding for the genetic observations presented.

1. The authors propose that Rad55-Rad57 stabilize Rad51 filaments independently of the protection against Srs2. Experimental evidence (such as ChIP or fluorescent microscopy) to support this model is warranted.

2. As suggested in the discussion, it would greatly improve the mechanistic insight of the study to test the model that Rad55-Rad57 directly prevents TLS polymerase recruitment (such as ChIP or fluorescent microscopy).

3. The model figure would be clearer if the consequences of knocking out the individual pathways or both pathways were also indicated. The dotted arrows made it seem like the pathways were still active. Perhaps “x”s could be placed on top of the arrows to show that a particular pathway is blocked.

Reviewer #2: The manuscript entitled “Rad51 filaments assembled in the absence of the complex formed by the Rad51 paralogs Rad55 and Rad57 are outcompeted by translesion DNA polymerases on UV- induced ssDNA gaps” examined the role of Rad55/Rad57 in UV-induced HR and cellular tolerance to UV. While previous studies examined DSB repair of the rad55 and rad57 mutants, this study uncovered the crucial role of Rad55/Rad57 in gap repair. This is an important conclusion, and this manuscript should be publishable in PLoS Genet if appropriate revision is done.

Specific comments:

1. The two Tad51 paralogs facilitate the formation of Rad51 filaments. However, this study did not analyze the formation of Rad51 filaments following UV irradiation. The authors need to show the formation of Rad51 filaments in the rad55 or rad57 mutant comparing X-ray-induced DSBs and UV-induced gaps.

2. The title seems to be misleading to general readers. The rad55 and rad57 mutants display only modest UV sensitivity in the presence of functional translesion synthesis (TLS) DNA polymerases while the rad30 and rev3 mutants are hypersensitive to UV (Fig. 2A and B and Fig.3C). A straightforward interpretation is “Rad51 filaments assembled in the PRESENCE of the complex formed by the Rad51 paralogs Rad55 and Rad57 are outcompeted by translesion DNA polymerases on UV- induced ssDNA gaps”. The reviewer believes that the authors’ finding is very important even when the two Tad51 paralogs acts as a backup for the TLS-dependent rapid release from replication blockage at UV-damage. Please consider changing the title.

3. This manuscript is further improved if the authors show the fiber analysis, IdU labeling, UV irradiation, and CldU labeling, comparing wild type, rad55 mutant, rev3 mutant, and rad55/rev3 double mutant, if possible.

4. Please show the survival data of rad55/srs2 double mutant. Is this mutant lethal?

5. Page 6 line 131: “Rad55-Rad57 complex is essential for UV-induced HR, but only accessory for IR-induced HR”. Many readers know little about yeast genetic studies, and the authors need to show the data of rad51 mutant in Fig. 1D and Fig. 2 and also the data of rad51/rev3 double mutant in Fig. 2.

6. Page 14 line 329: “UV-induced lesions can be channeled from HR to TLS”. It is formally possible that a small portion of the UV-induced gaps are converted to DSBs that trigger a substantial fraction of the heteroallelic HR events (Fig. 1C). Moreover, unrepaired DSBs, but not remaining gaps, lead to cell death (Fig. 1B and D). Fig. 2C argues against this idea by showing that loss of Rad55 increased TLS-mediated mutagenesis and that Rad55 and TLS share the same substrate, single-strand gaps. Fig. 2C appears to be the only data that suggests “channeled from HR to TLS” when functional Rad55 is absent. To show Rad55 and TLS share the same substrate, single-strand gaps, the authors need to add the data of rev3 single mutant and rad55/rev3 double mutants to Fig. 2C.

Reviewer #3: My review is uploaded as an attachment

**Have all data underlying the figures and results presented in the manuscript been provided?**

Reviewer #1: Yes

Reviewer #2: Yes

Reviewer #3: Yes

PLOS authors have the option to publish the peer review history of their article (what does this mean?). If published, this will include your full peer review and any attached files.

Reviewer #1: No

Reviewer #2: No

Reviewer #3: **Yes: **JeanSeb Hoffmann

---

## [Decision Letter · Decision Letter 1]

26 Jan 2023

Dear Eric,

We are pleased to inform you that your manuscript entitled "Rad51 filaments assembled in the absence of the complex formed by the Rad51 paralogs Rad55 and Rad57 are outcompeted by translesion DNA polymerases on UV-induced ssDNA gaps" has been editorially accepted for publication in PLOS Genetics. Congratulations! The reviewers appreciate the effort that went into the revised manuscript and all agree that addition of RPA foci data provide strong support for the model. This is an important study because it addresses recombination initiated at ssDNA gaps, which are likely the relevant lesion for most spontaneous recombination reactions in mitotic cells. 

Yours sincerely,

Lorraine S. Symington

Academic Editor

PLOS Genetics

Gregory P. Copenhaver

Editor-in-Chief

PLOS Genetics

Comments from the reviewers (if applicable):

Reviewer's Responses to Questions

**Comments to the Authors:**

Reviewer #1: The authors have done a thoughtful job in addressing my concerns. The RPA foci contributed a lot to the revision of the paper.

Reviewer #2: i would like to support the publication of this work in PLoS Genet.

Reviewer #3: The authors have improved their manuscript and have taken into account all my comments/concerns.

I appreciate the authors' effort in improving this manuscript and I consider the manuscript to now be suitable for publication.

**Have all data underlying the figures and results presented in the manuscript been provided?**

Reviewer #1: Yes

Reviewer #2: Yes

Reviewer #3: Yes

PLOS authors have the option to publish the peer review history of their article (what does this mean?). If published, this will include your full peer review and any attached files.

Reviewer #1: No

Reviewer #2: No

Reviewer #3: **Yes: **Dr. Jean-Sébastien Hoffmann

**Data Deposition**

http://datadryad.org/submit?journalID=pgenetics&manu=PGENETICS-D-22-00336R1

**Press Queries**

---

## [Editor Report · Acceptance letter]

2 Feb 2023

PGENETICS-D-22-00336R1 

Rad51 filaments assembled in the absence of the complex formed by the Rad51 paralogs Rad55 and Rad57 are outcompeted by translesion DNA polymerases on UV-induced ssDNA gaps 

Dear Dr Coïc, 

We are pleased to inform you that your manuscript entitled "Rad51 filaments assembled in the absence of the complex formed by the Rad51 paralogs Rad55 and Rad57 are outcompeted by translesion DNA polymerases on UV-induced ssDNA gaps" has been formally accepted for publication in PLOS Genetics! Your manuscript is now with our production department and you will be notified of the publication date in due course.

With kind regards,

Anita Estes

PLOS Genetics

On behalf of:
